# Ultra-Broadband, Omnidirectional, High-Efficiency Metamaterial Absorber for Capturing Solar Energy

**DOI:** 10.3390/nano12193515

**Published:** 2022-10-08

**Authors:** Jing-Hao Wu, Yan-Long Meng, Yang Li, Yi Li, Yan-Song Li, Gui-Ming Pan, Juan Kang, Chun-Lian Zhan, Han Gao, Bo Hu, Shang-Zhong Jin

**Affiliations:** 1College of Optical and Electronic Technology, China Jiliang University, Hangzhou 310018, China; 2State Key Laboratory of Applied Optics, Changchun Institute of Optics, Fine Mechanics and Physics, Chinese Academy of Sciences, Changchun 130033, China; 3The Postdoctoral Center of the Department of Electronic Engineering, Fudan University, Shanghai 200433, China

**Keywords:** absorption, ultra-broadband, multilayer, nanostructure

## Abstract

In this study, we investigated an absorber based on a center-aligned tandem nanopillar array for ultra-broadband solar energy harvesting theoretically. A high-efficiency, omnidirectional absorber was obtained by introducing the center-aligned tandem nanopillar array embedded in an Al_2_O_3_ dielectric layer. The multi-coupling modes at different wavelengths were interpreted. The strong absorption can be adjusted by changing the radii and heights of nanopillars. According to the simulation results, the average absorptance of the absorber exceeded 94% in the wavelength range from 300 nm to 2000 nm. In addition, the high-efficiency absorption was insensitive to the incident angle and polarization state. The research not only proposed an absorber which possesses a huge potential value for application areas, such as thermal photovoltaic systems, infrared detection, and isotropic absorption sensors, but also pointed out a new way to design an absorber with high efficiency in an ultrabroad wavelength range.

## 1. Introduction

Absorbing solar energy sufficiently in the full spectrum range is essential for solar energy utilization. Because of the limited extinction coefficient of natural material and the reflection on the interface, a perfect absorption for the full solar spectrum is hard to realize. In virtue of the advantages of manipulating electromagnetic waves, metamaterials have attracted more and more attention. Until now, no research has been done to explore its potential application fields [1,2,3,4,5,6,7,8,9]. For perovskite solar cells, the use of micro-nanostructures to improve the management of light entering the device is an effective way to improve conversion efficiency [10,11]. Many metamaterial structures have also been proposed to improve the absorbing ability of absorbers [12,13,14,15,16]. Landy et al. proposed the use of metamaterials to obtain an ultra-high absorption in a narrow frequency range [12]. Hubarevich et al. proposed an insulator-metal-insulator-metal (IMIM) nanostructure that helped the absorber achieve an average absorption of 82.5% in the range from 300 nm to 752 nm [17]. Zhu et al. proposed a stacked micro-nanostructure that achieved an average absorption exceeding 80% in the range from 400 nm to 700 nm [18]. Lei et al. also proposed an absorber based on a periodic array, which was composed of tandem titanium and silica nanocubes, for realizing an average absorption of 97% in the wavelength range from 354 nm to 1066 nm [19]. Near-perfect absorption in broadband was also proposed and investigated by using metal-insulator-metal (MIM) structures [20,21,22]. The enhancement of the absorption ability of such devices is attributed to a strong enhancement of inner electric field intensity which leads to an obvious improvement of absorption in the metal. However, the near-perfect absorptions reported in the literature are limited to a narrow band. Broadband absorption can also be achieved in micro-nanostructures, in which the strong absorption is realized by Mie resonances [23,24,25]. In order to ensure the absorbers can work normally at a very high temperature, the materials selected need to have not only a high absorptive coefficient in a wide wavelength range, but also a high stability. Since metal Ti and its compounds have a high melting point and excellent optical loss characteristics, presenting many advantages for the broadband absorption of light, Ti and its compounds are widely considered as candidates for application in absorbers working at a high temperature. Yu et al. used TiN to achieve an average absorption exceeding 90% in the range from 360 nm to 1624 nm [26]. Li et al. used a cross-shaped TiN structure that could achieve more than 90% light absorption in the wavelength range from 415.6 nm to 1597.39 nm [27]. All these researches showed that high light absorption efficiency could be obtained in a certain spectral range using Ti and its compounds. Though the wavelength range reported in these researches possessed the dominant radiation energy, the energy in the mid-infrared and far-infrared range was still valuable for enhancing the utilization.

This paper presents a high-efficiency absorption structure using refractory ceramic and metallic materials, such as titanium, aluminum oxide, silicon oxide, and titanium nitride. The structure is composed of a metallic nanopillars array embedded in an Al_2_O_3_ dielectric layer, which is set on MIM multilayers. On top of each nanopillar is a well-designed circular hollowing, of which the radius is different from that of Ti nanopillar. The well-designed structure exhibits nearly perfect absorption in the wavelength range from 300 nm to 2000 nm. The effects of hollowing depth and metal nanopillar height on the absorption performance of the structure are investigated in detail. The evolution of the absorption as the incident angle and polarization states change is also discussed. Finally, the capability of the device for capturing solar energy has been demonstrated [19,28,29,30,31].

## 2. Model Design and Simulation

The structure of the proposed broadband absorber is shown in Figure 1. As shown in Figure 1a, the base of the absorber is a thick TiN layer. In order to ensure that light is only absorbed and reflected when it is incident on the absorber, the thickness of TiN is set to 400 nm, which is thick enough to ensure that light cannot pass through the absorber. Above the thick TiN are SiO_2_ and a thin Ti layer. The Ti, SiO_2_, and TiN base consist of a simple MIM structure. A center-aligned air and Ti nanopillar array surrounded by an Al_2_O_3_ dielectric layer is set on top of the MIM structure. The aligned center favors the practical fabrication of such a structure. In Figure 1b, H_1_~H_5_ represents the thickness of the hollow air cylinder, Ti nanopillar, Ti thin film, SiO_2_, and TiN, respectively.

The initial structural parameters of the absorber without nanopillars (except for nanopillars) used for simulation, are listed in Table 1. The research method used in this study is the finite difference in time domain (FDTD) method. The background index is set to 1. The refractive index of SiO_2_ is set to 1.48. In the FDTD simulation, the boundary conditions in the X-directions and Y-directions are set to periodic boundary conditions, and the perfectly matched layer is set in the Z-direction to eliminate boundary scattering. The lengths of one period in x and y directions are set to 800 nm and 650 nm. There are five center-aligned stacked nanopillar structures in one periodic cell. The distance between the surrounding nanopillars in x and y directions is 440 nm and 280 nm, respectively. That means the nanopillars are arranged in rows and columns with different spacing. The light source used in the simulation is a plane wave, the wavelength of which ranges from 300 nm to 2000 nm. The light propagates along the forward Z-axis direction. Since we use a thick enough TiN as the substrate of the structure to ensure that the light cannot be transmitted, the absorptance of the absorber can be obtained according to the equation A = 1 − R. The data of reflectance R is obtained by a power monitor above the structure. The optical constants of materials used in the simulation are plotted in Figure 2, which are quoted from PALIK’s database [32,33,34,35,36]. All the materials chosen in the design have a high melting point, which indicates a great potential value for applications in various fields. The utilization of non-precious metals is also beneficial for reducing manufacturing costs. In the manufacture of micro-nanostructures, Hitoshi et al. [37] and Yi et al. [38] successfully prepared grating micro-nanostructures and pore micro-nanostructures on the W surface, respectively. Seo et al. explored using Laser Interference Lithography to prepare a variety of micro-nanostructures in optics [39], while Fan et al. used a picosecond laser beam to fabricate porous coral structures on the surface of Cu and achieved an average absorption rate of more than 90% in the range of 250 nm to 2500 nm [40]; these studies provide a reference for the absorber manufacturing proposed in this paper.

## 3. Results and Discussion

In order to clarify the function of each component utilized in the designed structure, the absorptive spectra in the four principal structures are compared in Figure 3. As shown in Figure 3a, the absorption gradually decreases as wavelength increases in the traditional MIM structure. This can be improved by adding a dielectric layer Al_2_O_3_ on top of the MIM structure to form an IMIM structure. Introducing Al_2_O_3_ not only improves the absorption in the mid-infrared range, but also enhances the absorption in the visible range. As a result, maximum absorption in the modified MIM structure can reach 100 percent. The reason for the improved absorption is that the addition of an insulating layer changes the plasmon resonance modes of the planar structure. The insulator with a higher reflective index causes the plasmonic resonant wavelength to shift to a longer wavelength [17]. Although the overall absorption efficiency of the IMIM structure is higher than that of the MIM structure, there still is a waste of energy absorption in the wavelength range from 800 nm to 1500 nm. It is also well known that the nanopillar array has many advantages in favor of high absorption, such as long propagation length of light and large surface area. The absorption coefficient, α, is related to the extinction coefficient, *k*, by the following formula [41]:(1)$$\alpha =\frac{4\pi k}{\lambda }$$
where *λ* is the wavelength. As can be seen from Figure 2, the extinction coefficient *k* of the metal Ti has a large value in the visible and near-infrared wavelength ranges, thus ensuring that Ti has a high optical loss in this range. In addition, Ti has a melting point of up to 1660 °C, which also guarantees that it can adapt to higher working conditions. Due to its good optical and physical properties, it is often chosen as a material for broadband absorbers. The absorptive spectra of IMIM structure with air and Ti nanopillar arrays are plotted in Figure 3b. The radius of the air and Ti nanopillar is 120 nm and 95 nm, respectively. As shown, both kinds of nanopillar arrays have good light absorption efficiency in the 300 nm to 600 nm band range. However, the structure with air nanopillars shows a contrary evolution trend as the wavelength increases from 600 nm to 2000 nm. As a result, the light absorption efficiency of the air nanopillars structure is superior in the wavelength range of 600 nm to 1600 nm and weaker in the wavelength range of 1600 nm to 2000 nm than that of Ti nanopillars.

To further understand the effect of the two nanostructures on light absorption, we kept the central nanopillar position unchanged, defined the center position of the upper and lower nanopillar as L_1_ (air nanopillar) and L_2_ (Ti nanopillar), and changed the parameters L_1_ and L_2_ to study the effect on light absorption. As shown in Figure 4, (a) the absorption spectrum is obtained by changing the L_1_ distance in the air nanopillar, and (b) the absorption spectrum is obtained by changing the L_2_ distance in the Ti nanopillar. As can be seen from Figure 4a, for the air nanopillar array’s structure, changing the distance L_1_, the main impact band is between 700 nm and 800 nm, and as the distance between the center of the circle becomes longer, the peak of the absorption peak appears redshift, and the peak decreases. This is because the height of the air cylinder is close to *λ*/4, at which point a Fabry-Perot resonance is formed [42], which allows the cavity (Fabry Perot-like), more excavated by the grooves formed on the surface of the structure, to also participate in the absorption of visible light [42,43,44]. When the distance between the two nanopillars increases, the resonance effect between the cavities weakens, resulting in a decrease in peaks. As can be seen from Figure 4b, for Ti nanopillar array’s structure, changing the distance L_2_, the main affected band is between 800 nm and 1600 nm, and as the center distance of the circle becomes longer, the absorption rate of this band shows a downward trend. This is because Ti has good light loss characteristics in this band, and, as the distance increases, the resonance effect between Ti nanopillars weakens, resulting in decreased absorption.

To further improve the absorption in the full wavelength range, we tried to stack the air and Ti nanopillar to complement the drawbacks of each other. The height ratio of air and Ti in a single stacked nanopillar is 1:1. The radius of the lower Ti nanopillar is 95 nm. The absorptance curves of the devices with different radii of the upper air nanopillar are plotted in Figure 5a. The simulation periods x and y remained constant, and the area filling rate of the air nanopillar increased from 14.8% to 43.5%. Here filling ratio is defined as [45]:(2)$$F=\frac{5\times \pi \times r^{2}}{S}$$
where *r* is the radius of air nanopillar, *S* is the area of the simulation cycle. It is straightforward that the absorptances of all the devices with different air nanopillar radii in the infrared are enhanced, exceeding 0.93 in the total range from 1000 nm to 2000 nm. As the radius of air nanopillar increases, the absorption also presents an increasing trend in the infrared range except for the wavelength range near 2000 nm. A similar trend can be observed from 400 nm to 600 nm. However, there is a considerable fluctuation in the range from 600 nm to 800 nm. In this range, the absorption varies from a peak to a valley, forming a complete contrast. In order to clarify the reason behind this, the electric field distributions at 780 nm are carried out on the horizontal cross-section of these devices, as shown in Figure 5b–d. It can be seen from Figure 5b that the area with a strong electric field is mainly in the middle outer part of the upper and lower circular hollowing, and on both sides of the middle circular hollowing. When the radius is 95 nm, the strong electric field region between the upper and lower circular hollowing further expands and moves to the middle region of the upper and lower circular hollowing, and the strong electric field areas on both sides of the middle circular hollowing also increase. When the radius reaches 75 nm, the strong electric field area is mainly concentrated in the middle area of the upper and lower circular hollowing and on both sides of the middle circular hollowing. Additionally, regarding the reduction of the radius, the range of the strong electric field area, and the electric field value of the strong electric field area, increase. This indicates that the surface of the structure can form a strong resonant cavity at this wavelength. Combined with the slow-wave effect, this effect lengthens the process of interaction between light and matter, and more solar energy is absorbed. According to the slow-wave effect theory and the effective medium theory [46], the expression for slow-wave effect is [47]:(3)$$\begin{array}{l} \lambda_{P}=2W\sqrt{\varepsilon_{⊥}} \end{array}$$
(4)$$\frac{1}{\varepsilon_{⊥}}=\frac{f}{\varepsilon_{m}\left(\omega \right)}+\frac{1-f}{\varepsilon_{d}\left(\omega \right)}$$
where *f* is the filling rate, obtained by Equation (2), and *ε_m_* and *ε_d_* are the relative permittivity of Ti and Al_2_O_3_, respectively. From Equations (2)–(4), taking *W* = 200 nm and *f_Ti_* = 0.169 as an example, we can obtain that *λ_P_* is equal to 780 nm, consistent with the analytical wavelength. This means that there are indeed slow-wave effects happening in that band that also result in strong light absorption.

After studying the effect of air-nanopillar radius on the absorber, the height ratio of Ti to air nanopillar is investigated. The radii of air and Ti nanopillars and their total height are fixed to 120 nm, 95 nm, and 200 nm, respectively. As shown in Figure 6, five different height ratios were designed. It is straightforward that there is a dramatic variation in the absorptive spectrum as the ratio changes. Though there is a slight decline in the absorptance from 300 nm to 600 nm as the height ratios of Ti to air nanopillar decrease, the total absorptions of the absorbers with lower height ratios still present better absorptive performances. There is a remarkable improvement in the absorption as the height ratios exceed 1:1. As can be seen in Figure 6, the improvement mainly occurs in two wavelength ranges. The noticeable wavelengths in the two ranges are 752 nm and 1232 nm.

To address the mechanism, the distribution of power density at 752 nm is analyzed first. The absorbed power density in the material is given by [48]:(5)Pabs=ωε08πIm[ε]|E|
where Im[*ε*] is the imaginary part of the complex permittivity of the medium, |E| is the amplitude of the electric field in the medium, *ω* is the angular frequency, and *ε*_0_ is the vacuum permittivity. The absorbed power density in the cross-section of the absorber, with various height ratios of Ti to air nanopillar, is shown in Figure 7. It is obvious that most of the absorption in this structure occurs around the Ti nanopillar under different height ratios. Along with the height decrease of the Ti nanopillar, the absorbed power density shows an increasing trend. The maximum absorbed power density reaches 3.6 × 10^19^ W/m^3^ and 4.4 × 10^19^ W/m^3^, as the volume ratios between Ti nanopillar and air cylinder are 1:2 and 1:3, respectively. Accordingly, the absorptance in 752 nm reaches 98.8% and 99.1%, respectively. When the height of the Ti nanopillar is greater than that of the air cylinder, the absorbed power density begins to decrease. That results in a poor absorptance at 752 nm. The variation in the absorptance as volume ratio changes is attributed to the electric fields coupling between the air cylinder and the surface of the Ti cylinder, i.e., the field generated by stimulated surface plasmon. The change in coupling results demonstrates a difference in the reflectivity of various nano-patterned layers, which leads to a change in the light absorptive efficiency.

The distributions of the electric field in the xz and xy planes at 1232 nm were also analyzed by using a field monitor. Figure 8a–e shows the electric field distribution in the xz plane as well as the position of the monitor. It can be seen that there is always a strong electric field around the Ti nanopillar regardless of the height ratio. Moreover, it also should be noted that most of the electric field is concentrated around the Ti nanopillar. As the height ratio of Ti to air nanopillar is 3:1, the localization effect is so strong that there is almost no electric field passing through Al_2_O_3_. Then the localized surface plasmonic resonant (LSPR) absorption of Ti nanopillar dominates the total absorption. That leads to weak absorption of middle Ti thin film directly. As the height ratio decreases, the localization effect of the electric field brought by the Ti nanopillar is weakened. More electric field energy reaches the middle Ti film’s surface. Surface plasmonic resonant (SPR) absorption becomes increasingly important. As can be seen in Figure 8e, there is a strong coupling between LSPR of Ti nanopillar and SPR of middle Ti thin film. Figure 8f–j shows the electric field distribution in the XY plane. It can be seen that when the height of the Ti nanopillar is greater than the air nanopillar, the electric field with higher intensity is mainly concentrated around the Ti nanopillar. There is a strong coupling of electric fields among the Ti nanopillars. As the height of Ti nanopillar decreases, the electric field begins to shrink on the corner of Ti nanopillar. The electric field coupling degrades. When the height ratio of the Ti to air nanopillar is less than 1:1, the strong coupling between Ti nanopillar disappears and is replaced by a strong electric field in the whole plane. The analysis of electric field distribution manifests that the improvement of absorption in the range between 1000 nm to 1600 nm derives from the resonant coupling enhancement between LSPR of Ti nanopillar and SPR of middle Ti thin film.

In order to further evaluate the role of Ti thin film in the absorption, the absorptive spectra of the absorber with different intermediate thin films are plotted in Figure 9a. Because of the higher extinction coefficient, the absorber using W as an intermediate thin film presents a better absorptive capability in the visible range. However, the low refractive index of Ti is more suitable to restrict the electric field in the dielectric layer, forming a higher localized electric field resonant enhancement. As a result, the absorptive capability of absorber with W thin film is worse than that of absorber with Ti thin film. As to the absorber with Au and Ag thin films, the overall low extinction coefficient of Au and Ag results in a worse absorptive capability than the device with Ti thin film. According to the results, Ti is more satisfactory for this structure. The effect of Ti layer thickness on the absorption is also shown in Figure 9b. From Figure 9b, it can be seen that without the planar Ti layer, there is a substantial decrease in overall absorption. It is well known that Ti possesses strong energy loss in the visible and near-infrared range, and a large mismatch of refractive index occurs in comparison with the dielectric layer used in the absorber. These merits help Ti thin film to play an important role in this structure. The absorbers with 12 nm and 14 nm thick Ti thin film show a similar absorptance. However, there is a distinct derivation on the absorptive spectrum as the thickness of Ti increases to 16 nm. It indicates that there is a thickness saturation value when the planar Ti layer plays the role of absorption, and increasing the thickness after reaching a certain value does not have a significant effect on the improvement of the overall light absorption efficiency of the absorber.

According to the discussion above, the optimized structural parameters are decided and listed in Table 2. R_1_ and R_2_ stand for the radius of the air and Ti nanopillars. The absorptive and reflective performance of the absorber with optimized structure parameters is shown in Figure 10. Where absorption efficiency can be expressed as [49]:(6)$$\alpha =\frac{\int_{\lambda_{\min}}^{\lambda_{\max}}\left(1-R(\lambda )-T\left(\lambda \right)\right)\times S_{AM1.5}\left(\lambda \right)d\lambda }{\int_{\lambda_{\min}}^{\lambda_{\max}}S_{AM1.5}\left(\lambda \right)d\lambda }$$
in the above equation, S_AM1.5_(*λ*) is the energy corresponding to the standard solar spectral data, R(*λ*) is the reflectivity, and T(*λ*) is the transmittance. It can be seen that the average absorptance exceeds 94.3% in the wavelength range of 300 nm to 2000 nm. The average absorption of the absorber can be expressed as [50]:(7)$$\bar{A}=\frac{\int_{\lambda_{\min}}^{\lambda_{\max}}A\left(\lambda \right)d\lambda }{\lambda_{\max}-\lambda_{\min}}$$
where *λ*_max_ and *λ*_min_ denote the maximum and minimum wavelengths of the absorption spectrum, respectively.

Though there are two absorption valleys in the visible range where the absorber performs weakly, the overall average absorptance in the visible range can still reach 94.3%. Because of the weak extinction coefficient, the absorption in the infrared range usually tends to decrease. However, even then, the overall absorption in the infrared range still presents better results than that in the ultra-violet and visible ranges. The average absorptance in the infrared range can reach 96.6%, and it is as high as 99% evenly in the range of 1650 nm to 1800 nm. The light absorption efficiency exhibited by this structure conforms to the expectation of broadband and high absorption, indicating that stacking the two kinds of nanopillar in a traditional IMIM structure can effectively improve the overall light absorption efficiency. In addition, as shown in Table 3, we cited some examples of ultra-broadband high-absorption devices that have used refractory metals in the past [19,28,29,30,31]. By comparison, we can conclude that our absorber has a better absorption effect.

To investigate the dependence of the absorber on the angle, we simulated the absorption evolution as the incident angle and polarization state changed. As shown in Figure 11a, there is a slight decrease in absorption as the incident angle varies from 0° to 55°. However, the absorptance is still maintaining above 90%, and it is notable that the absorptance keeps above 95% in the range from 1550 nm to 1650 nm. As the polarization angle of the light changes from 0° to 90°, there is almost no changes in the absorption, as shown in Figure 11b. These results show that the proposed absorber is angle-insensitive and polarization-insensitive due to the well-designed symmetrical structure. That also allows the absorber to perform well when it is utilized in the photothermal system and optic camouflage.

In order to verify the final absorption results of the absorber, this paper calculated the absorbed solar spectral power density distribution based on the standard solar radiation spectrum AM1.5, which can be expressed as [51]:(8)$$A=\int_{300}^{2000}a\left(\lambda \right)S_{AM1.5}\left(\lambda \right)d\lambda$$
where a(*λ*) represents the absorption spectrum of the structure, S_AM1.5_ refers to the AM1.5 solar spectrum. As shown in Figure 12a, the black curve represents the standard solar radiation spectrum of AM1.5, and the red curve represents the absorption spectrum of the absorber. It can be seen that the absorber maintains a strong absorption property in the range of 300 nm to 2000 nm, and can show high-efficiency absorption in the wavelength range beyond 800 nm. As shown in Figure 12b, the red area indicates the absorbed energy of the absorber, and the orange area indicates the energy lost by the absorber. It can be seen that the energy lost by the absorber is mainly concentrated in the wavelength range of 400 nm to 800 nm, but the energy lost is not much. Therefore, it can be shown that the absorber proposed in this paper has certain advantages in resonant absorption in the wavelength range from 300 nm to 2000 nm, and also shows that the absorber plays a good effect in energy absorption and solar photothermal system.

Finally, considering that absorbers also emit spontaneous radiation as well as solar radiation, and based on Kirchhoff’s law, under the condition of thermodynamic equilibrium, the ratio of monochromatic radiation emission of different objects to the same wavelength is equal to the ratio of monochromatic absorption, and is equal to the degree of monochromatic radiation of blackbody to the same wavelength at that temperature [52], we calculated the photothermal conversion efficiency of the absorption. The vertical emissivity of the absorber in the 300 nm to 2000 nm band can be calculated by the following expression [53]:(9)$$\varepsilon =\frac{\int_{300}^{2000}\varepsilon_{\lambda }I_{B}\left(\lambda ,T\right)d\lambda }{\int_{300}^{2000}I_{B}\left(\lambda ,T\right)d\lambda }$$
where I_B_ (*λ*, T) is the blackbody radiation at the working temperature T [54]. In addition, the blackbody radiation can be expressed as [55]:(10)$$I_{B}\left(\lambda ,T\right)=\frac{2\pi {hc}^{2}}{\lambda^{5}}\times \frac{1}{E^{hc/\lambda kT}-1}$$
where *h* denotes the Planck constant, *c* indicates the velocity of electromagnetic wave in vacuum, and *k* is the Boltzmann constant the solar absorber can be calculated using the following expression [56]:(11)$$\eta =\frac{\alpha G-\varepsilon \left(\sigma T^{4}-\sigma T_{S}{}^{4}\right)}{G}$$
where G refers to the total incident irradiation and can be expressed as G = C × 1000 W/m^2^ [57], C is concentration factor, α can be obtained from Equation (6), *σ* is Stefan-Boltzmann constant, set the operating temperature T_s_ to 0°C. Figure 13 shows the photothermal conversion efficiency of the absorber at different temperatures and the conception factor. We selected a temperature range of 400 K to 1000 K, and as the proportion factor increases, the photothermal conversion efficiency of the device also increases, and when the conversion factor is 1000, the overall conversion efficiency can be more than 90%. As the operating temperature increases, it leads to higher Carnot efficiency, but at the same time leads to a sharp increase in radiation losses [55].

## 4. Conclusions

In summary, an ultra-broadband, high-efficiency absorber for solar energy capturing was demonstrated by simulation. The intrinsic mechanism was also investigated. The results demonstrate that the overall absorption efficiency of the absorber can be improved by adjusting the coupling of the electric field, which is realized by adjusting the structural parameters of the air cylinder and Ti nanopillar surrounded by Al_2_O_3_. Finally, the composite absorber proposed in this paper exhibits an average absorption rate of more than 94.3% in an ultra-broad wavelength range from 300 nm to 2000 nm. For such a structure, it is extremely important to adjust electric field resonant coupling between the top composite structure and the bottom MIM structure by choosing the appropriate ratio of skeleton depth to metal nanopillar thickness to achieve near-ideal absorption at ultra-wide wavelengths. The absorber is also insensitive to light incidence angle and polarization states. Since all the selected materials, such as Ti, Al_2_O_3_, SiO_2_, and TiN, are high-temperature resistant, that makes the absorber adaptable to higher temperature environments and could reduce manufacturing cost in practical applications. Since the absorber has a good photothermal conversion efficiency, it can be used as an absorber in thermal photovoltaic systems [8,58]. In addition, the absorber has the characteristics of insensitivity to the angle of light incidence and polarization angle, so it can also be applied to isotropic absorption sensors [12,59].

## Figures and Tables

**Figure 1 nanomaterials-12-03515-f001:**
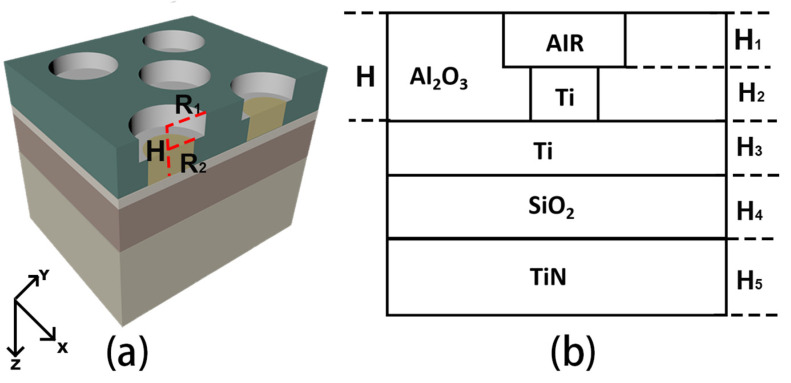
(**a**) The structure of the designed absorber. (**b**) The sketch of the absorber’s cross-section.

**Figure 2 nanomaterials-12-03515-f002:**
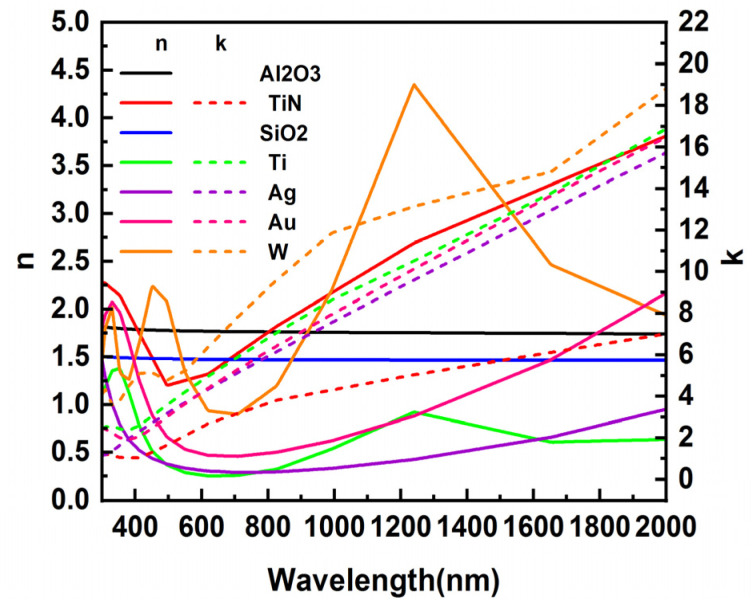
The refractive index and extinction coefficient of materials used in the simulation.

**Figure 3 nanomaterials-12-03515-f003:**
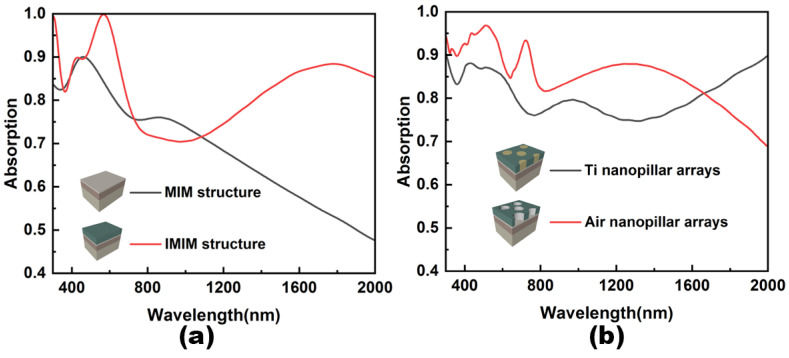
Absorptive spectra of different structures. (**a**) Absorptive spectra of MIM structure and IMIM structure. (**b**) Absorptive spectra of IMIM structure with Ti and air nanopillar arrays.

**Figure 4 nanomaterials-12-03515-f004:**
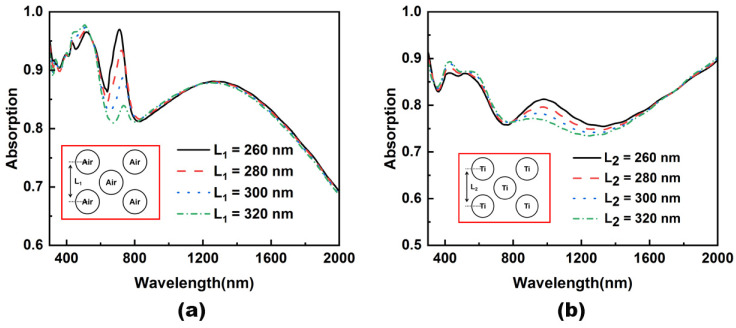
Absorptive spectra change of absorbers with air (**a**) and titanium (**b**) nanopillar arrays as the center distance between nanopillars is changed.

**Figure 5 nanomaterials-12-03515-f005:**
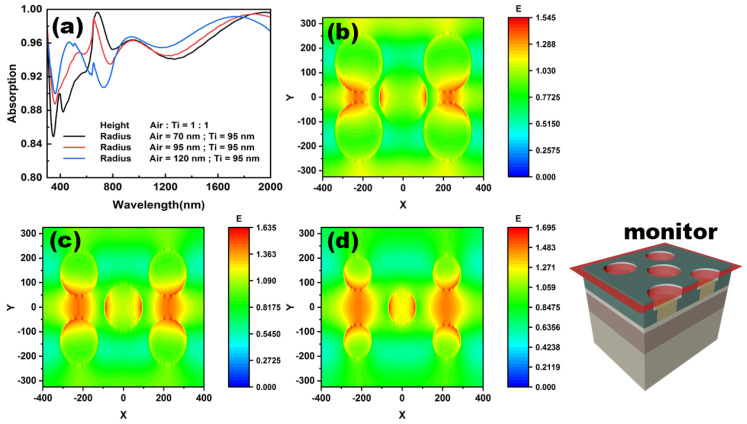
The absorptive spectra (**a**) and electric field distributions (**b**–**d**) in the same horizontal cross-section of the absorbers with different radii of air nanopillar, 120 nm (**b**), 95 nm (**c**), 70 nm (**d**). The height ratio of air to Ti nanopillar keeps 1:1.

**Figure 6 nanomaterials-12-03515-f006:**
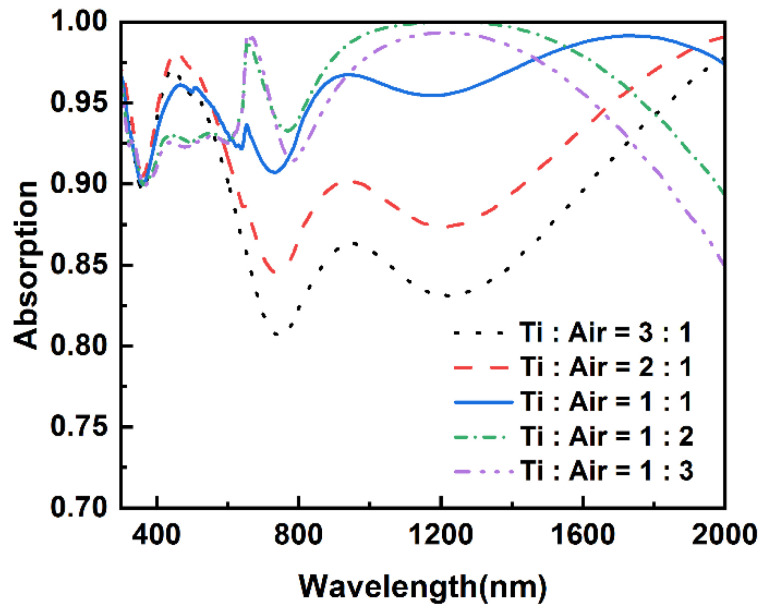
Absorptive spectra of absorber with different height ratios of Ti to air nanopillar, of which the radii are fixed to 95 nm and 120 nm.

**Figure 7 nanomaterials-12-03515-f007:**
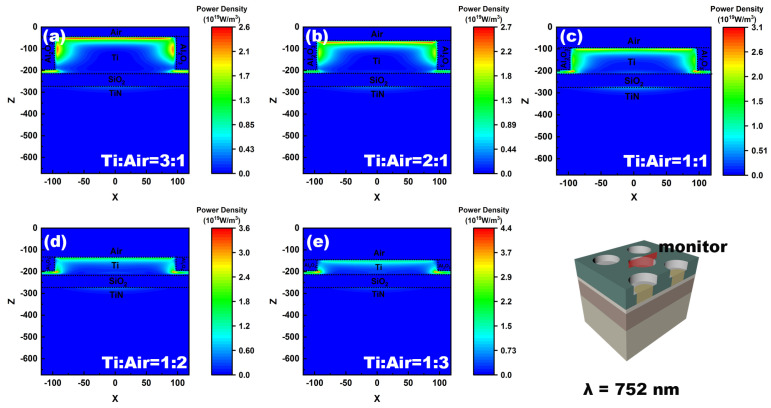
The distributions of absorbed power density in the vertical cross-section at the wavelength of 752 nm as the height ratio of Ti to air nanopillar are 3:1 (**a**), 2:1 (**b**), 1:1 (**c**), 1:2 (**d**), and 1:3 (**e**), respectively. The total height is fixed to 200 nm. The diameters of Ti and air nanopillars are fixed to 95 nm and 120 nm. The last subfigure shows the monitor position.

**Figure 8 nanomaterials-12-03515-f008:**
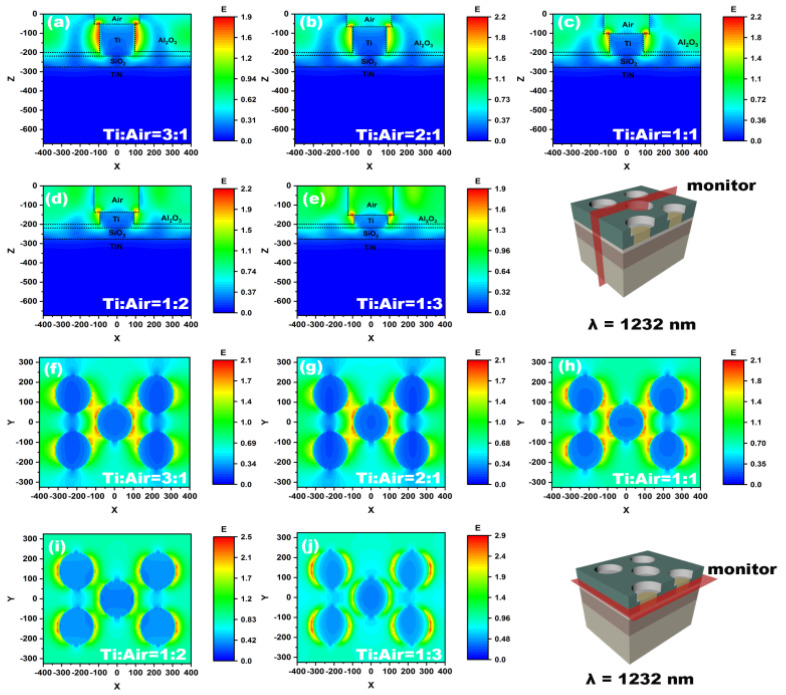
Distribution of electric field in the XZ cross-section (**a**–**e**) and XY cross-section (**f**–**j**) of absorber with different volume ratios at 1232 nm. The monitor in XY plane is set at the interface between Al_2_O_3_ and Ti thin film.

**Figure 9 nanomaterials-12-03515-f009:**
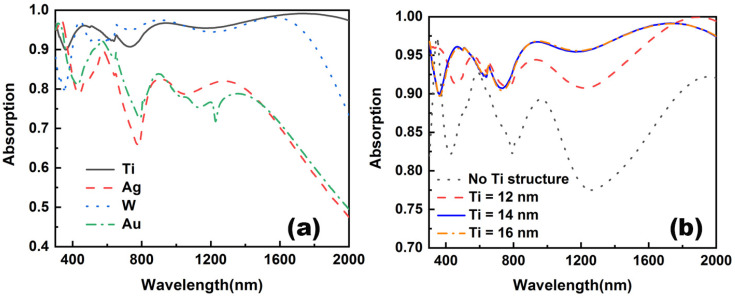
Absorption spectra of different cases of Ti layers. (**a**) Comparison of absorption spectra of different metals. (**b**) Comparison of absorption spectra of different Ti thicknesses.

**Figure 10 nanomaterials-12-03515-f010:**
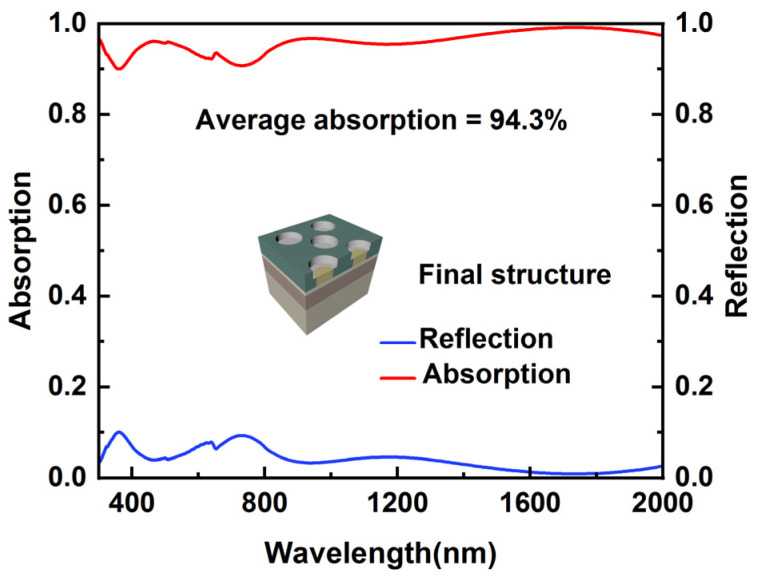
Absorptive and reflective spectra of the absorber at normal incidence.

**Figure 11 nanomaterials-12-03515-f011:**
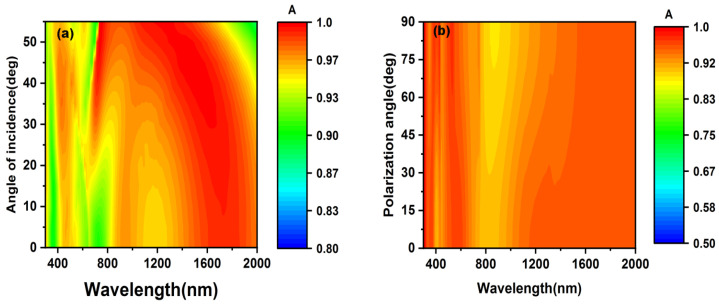
(**a**) The evolution of absorbers’ absorption as the incident angles increases from 0° to 55°. (**b**) The evolution of absorbers’ absorption as the polarization angles increases from 0° to 90°.

**Figure 12 nanomaterials-12-03515-f012:**
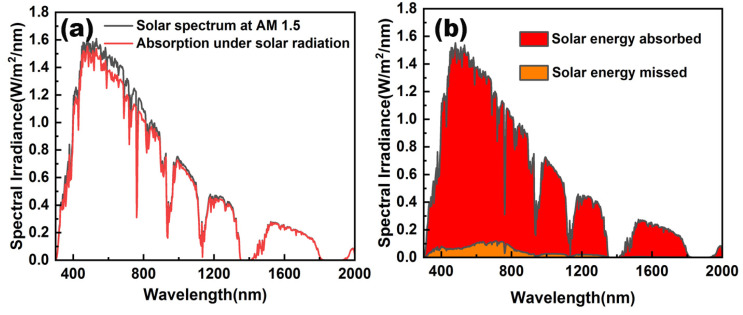
(**a**) Comparison of the AM 1.5 standard spectrum of solar radiation and the absorbed spectrum. (**b**) The energy absorbed and lost by the absorber under AM 1.5.

**Figure 13 nanomaterials-12-03515-f013:**
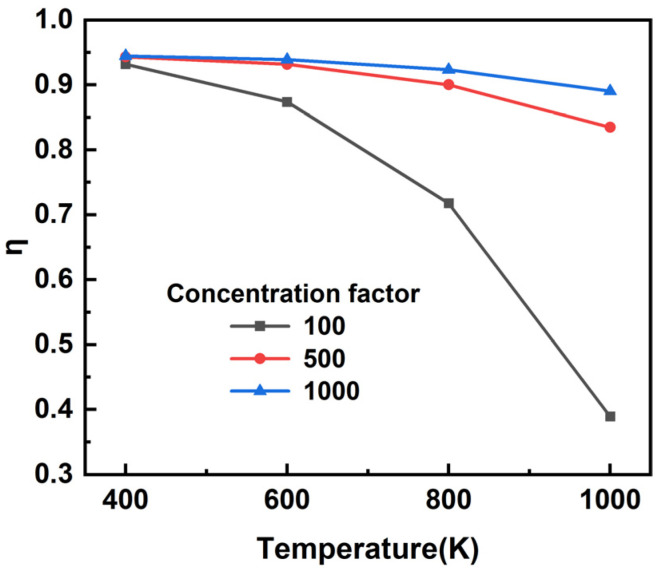
The Solar-thermal conversion efficiency for different solar irradiance at different temperatures.

**Table 1 nanomaterials-12-03515-t001:** Initial structural (except for nanopillars) parameters used in the simulation.

Layer	Radius (nm)	Height (nm)
Al_2_O_3_		H = 200
Ti		H_3_ = 14
SiO_2_		H_4_ = 60
TiN		H_5_ = 400

**Table 2 nanomaterials-12-03515-t002:** All optimized structural parameters used in the simulation.

Layer	Radius (nm)	Height (nm)
Al_2_O_3_		H = 200
Air	R_1_ = 120	H_1_ = 100
Ti	R_2_ = 95	H_2_ = 100
Ti		H_3_ = 14
SiO_2_		H_4_ = 60
TiN		H_5_ = 400

**Table 3 nanomaterials-12-03515-t003:** Comparison between the different absorber designs proposed in previous studies.

Refs.	Surface Structure	Bandwidth (Absorption > 90%)	Average Absorption in This Region
[19]	titanium-silica cubes	712 nm (354 nm–1066 nm)	97%
[28]	Titanium nitride ring-square	1000 nm (200 nm–1200 nm)	94.85%
[29]	titanium-silica cubes	1100 nm (405 nm–1505 nm)	95.14%
[30]	Circular-ring cell	1300 nm (300 nm–1600 nm)	95.77%
[31]	nanodisk–nanohole hybrid	400 nm (400 nm–800 nm)	>90%
Our work	nanopillar hybrid structure	1700 nm (300 nm–2000 nm)	94.3%

## Data Availability

The data presented in this study are available on request from the corresponding author.

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
