# Peer review of "Ultra-Broadband, Omnidirectional, High-Efficiency Metamaterial Absorber for Capturing Solar Energy"

_nanomaterials, 2022, doi:10.3390/nano12193515_

Round 1

Reviewer 1 Report

With this article, the authors propose an interesting solution for thermal energy harvesting, which is a critical concept to be addressed in perspective of decarbonization.

In the proposed article the authors simulated the performances, in terms average absorptance, of a novel ultra-broadband absorber. This latter is composed of a TiN-Si-Ti multilayer with a top layer of Ti and air nanopillars immersed in a dielectric matrix. The objective of the authors is to obtain a near-perfect absorber with a spectral absorption coefficient higher than 94.3% in an ultra-broad wavelength range from 300 nm to 2000 nm. This result, according to their simulations, has been achieved optimizing the dimensions (radius and height) of both air and Ti nanopillars, in order to adjust the coupling of electric field. The distribution of electric field is investigated to determine the prevalent absorption mechanism: Localised Surface Plasmonic Resonant (LSPR) absorption or Surface Plasmonic Resonant (SPR) absorption.

In my opinion this paper needs to be revised before the publication. In the following my suggestions and comments.

a) The meaning of “near-perfection” of the absorber must be clarified. 

b) In the abstract the authors state that the final purpose of the article is the energy harvesting. It would be appropriate to add some considerations about the radiative losses at high operating temperatures.

c) The simulated results, although promising, should be experimentally validated or, at least, the authors should provide a literature reference of the adopted model validation. 

d) Add the reference for the equations implemented in the Finite Difference in Time Domain method.

e) The authors, in the introduction, state that the high performance of their absorber is demonstrated: this statement should be accompanied by another reference with which to validate their results.

e) The author should improve the Conclusion section: about that I suggest adding specific possible applications of the proposed device.

f) Moderate English changes are needed, for example:

line 15) “in the wavelength range of (from) 300 nm to 2000 nm”;

75) The initial structural parameters of the absorber without nanopillars (except for nanopillars,) used for simulation are listed in Table 1.

85) of which the wavelength range is set (whose wavelength range/ the wavelength range of which)

118) Since Ti has a high optical loss in the visible and near-infrared wavelength range and a high melt (melting) point, it is often used in the broadband absorber. (This sentence needs to be clarified).

141-143) In order to clarify the reason behind this, the electric field distributions under (below) 780 nm at a horizontal cross-section of these devices are carried out, as shown in Fig. 4(b-d). 

201-203) It can be seen from the figure that there (is) always a strong electric field around the Ti nanopillar in despite of (regardless of) the height ratio. But (Moreover,) it also should be noticed that most of the electric field is concentrated around the Ti nanopillar.

230-232) However, the low refractive index of Ti is easier (more suitable) to restrict the electric field in the dielectric layer forming a higher localized electric field resonant enhancement.

Author Response

Dear  reviewer:

Thank you for your letter and the reviewers’ comments concerning our manuscript entitled “Ultra-broadband, Omnidirectional, Near-Perfect Absorber for Solar Energy Capturing” (ID #1907905). Those comments are all valuable and very helpful for revising and improving our paper, as well as the important guiding significance to our researches. We have studied comments carefully and have made corrections which we hope meet with approval. Our responses to the reviewers' comments are in the attachment.

Reviewer 2 Report

This article describes a device configuration of nanomaterials in order to improve the absorption over a very wide bandgap under a simulation point of view. The idea is novel and interesting but certain major issues should be improved before publication.

The main concern regarding the manuscript is the misunderstanding provoked by a title where the goal seems to be an optimization of the optical device fabrication instead of a theoretical optimization of an optical device driven by simulation approaches. This misunderstanding is not so severe in the abstract but, under my point of view, the abstract should be a bit clearer in the redaction to understand from the beginning that the only approach carried out is simulation.

Secondly, in a simulation article, a deep description of mathematical/physical modelling with several equations is expected. In this article the equations that describe the model employed are few. It is necessary to complete the article with all the equations used to confer more rigour to the article allowing readers to implement or eventually improve such simulations.

More references about the real or possible application of the devices here designed are necessary. Here are some References you can cite among others:

https://doi.org/10.1039/C8TC05461D

https://doi.org/10.1039/C5TA08743K

https://doi.org/10.1016/j.joule.2017.07.012

Finally, the effect of distance between nanopillars has not been fully analysed. Authors should explain such influence.

Author Response

(The authors gave the same response as above.)

Round 2

Reviewer 1 Report

The paper has been deeply revised, even if I think that a validation of the simulated results would be appropriate. The paper, in my opinion, can be published even if I suggest the authors to review some small English errors.

Author Response

Thank you for your letter and the reviewers’ comments concerning our manuscript entitled “Ultra-broadband, Omnidirectional, High-Efficiency Metamaterial Absorber for Capturing Solar Energy” (ID #1907905). Those comments are all valuable and very helpful for revising and improving our paper, as well as the important guiding significance to our researches. We have studied comments carefully and have made corrections which we hope meet with approval.Please see the attachment.

Reviewer 2 Report

Authors answered properly all my questions. The quality has improved significantly with by the introduction of the new Figure 4.

So, I can recommend the current corrected version for publication in Nanomaterials.

Author Response

(The authors gave the same response as above.)
